# Impact of Sn-doping on the optoelectronic properties of zinc oxide crystal: DFT approach

**Manoj Kumar**[1], **Purnendu Shekhar Pandey**[2], **Banoth Ravi**[3], **Bittu Kumar**[4], **S. V. S. Prasad**[1], **Rajesh Singh**[5], **Santosh Kumar Choudhary**[6], **Gyanendra Kumar Singh**[7] *

**1** MLR Institute of Technology, Hyderabad, India, **2** GL Bajaj Institute of Technology and Management, Greater Noida, India, **3** Indian Institute of Information Technology, Tiruchirappalli, India, **4** Koneru Lakshmaiah Education Foundation, Hyderabad, India, **5** Uttaranchal Institute of Technology, Uttaranchal University, Dehradun, India, **6** VNR Vignana Jyothi Institute of Engineering & Technology, Hyderabad, India, **7** Adama Science and Technology University, Adama, Ethiopia

* gyanendra.kumar@astu.edu.et

**Data Availability Statement:** All relevant data are within the paper and its Supporting Information files.

## Abstract

This study aims to provide a concise overview of the behavior exhibited by Sn-doped ZnO crystals using a computational technique known as density functional theory (DFT). The influence of Sn doping on the electronic, structural, and optical properties of ZnO have been explored. Specifically, the wavelength dependent refractive index, extinction coefficient, reflectance, and absorption coefficient, along with electronic band gap structure of the Sn doped ZnO has been examined and analyzed. In addition, X-ray diffraction (XRD) patterns have been obtained to investigate the structural characteristics of Sn-doped ZnO crystals with varying concentrations of Sn dopant atoms. The incorporation of tin (Sn) into zinc oxide (ZnO) has been observed to significantly impact the opto-electronic properties of the material. This effect can be attributed to the improved electronic band structure and optical characteristics resulting from the tin doping. Furthermore, the controllable structural and optical characteristics of tin-doped zinc oxide will facilitate the development of various light-sensitive devices. Moreover, the impact of Sn doping on the optoelectronic properties of ZnO is thoroughly investigated and documented.

## Introduction

In recent years, zinc oxide (ZnO) has received a lot of attention due to the fact that its unusual features can be used in numerous technologies, such as solar cells [1], sensors [2], transparent electrodes [3], and light emitting diodes (LED) [4, 5]. ZnO has been identified as a leading candidate for the development of high-performance opto-electronic devices due to its n-type wide-band gap material status (3.37 eV) and high exciton binding energy (60 meV) at ambient temperature [6, 7]. To be more specific, it has been looked at as a possible replacement for indium tin oxide (ITO) material, which is one of the most well-known TCOs due to its great optical transparency and low electrical resistivity [3, 8]. Furthermore, the ZnO coating allows visible light to pass through because of ZnO's large energy band gap (3.37 eV), which enables the devices to function in the blue-ultraviolet region [9–12]. Doping ZnO with metal atoms like Al, Ga, In, Sn, and Sb can boost its optical and electrical properties [13]. It is common

**Funding:** This work is supported by Adama Science and Technology University, Adama, Ethiopia. "The funders had role in study design and analysis.

**Competing interests:** The authors have declared that no competing interests exist.

knowledge that ZnO's low electrical conductivity prevents it from being used in a wide variety of potential TCO applications despite its great transparency in the visible area. To improve its electrical characteristics, ZnO is typically doped with a few carefully chosen elements. Numerous investigations have found that the electrical and optical properties of ZnO thin films can be enhanced by doping with elements from group III (Al, Ga, In) [5, 14, 15], group IV (Sn) [16–19] and group II (Mg) [20]. The Sn atom which is Group IV element, is often regarded as one of the suitable dopants since for substitution and interstitial doping [5, 21, 22] of ZnO, since it has atomic radius comparable to Zn atom [8, 22]. Dopants can occupy a wide variety of crystallographic positions within the host materials, and this can cause significant variation in the material's properties. However, the effect of tin (Sn) doping of ZnO crystal on its optical and structural characteristics have not been much explored to the author's best of knowledge. Moreover, it is challenging to experimentally analyze the differences in material properties related to different amount of dopants, making theoretical/computational calculation, such as density functional theory (DFT), a more practical method for investigating these phenomena. In particular, DFT calculations can aid in identifying the optimal amount of dopant to maximize Sn doped ZnO structure stability while maintaining desirable attributes. While investigating the optical and structural characteristics, all the possibilities of Sn interstitial doping of ZnO have been taken into consideration. Therefore, DFT has been utilized to look into the electronic band structures and its few optical properties such as absorption coefficient, refractive index and extinction coefficient which contribute to the unique optical capabilities of Sn-doped ZnO. Thus, the Sn-doped ZnO crystal's potential in optoelectronic applications has been assessed by demonstrating and explaining how to tune its optical and structural properties.

## Computational details

To further understand the Sn-doped ZnO's opto-electronic characteristics, we have performed DFT calculations using the Cambridge Sequential Total Energy Package (CASTEP) tool kit package of Material Studio [11]. The computations are carried out in Materials Studio using the generalized gradient approximation (GGA) and the Perdew, Burke, and Enzenhofer (PBE) method. In this work, hexagonal wurtzite crystal structure of ZnO have been considered with space group symmetry P63mc and primitive cell characteristics of a = b = 0.3250 nm, c = 0.5207 nm, α = β = 90˚, and γ = 120˚ [23]. ZnO crystals were modelled as 2×2×2 periodic super cells, with Sn atoms added as an interstitial dopant, and the resulting structures were used in the calculation. Fig 1(A) shows undoped ZnO supercell, Fig 1(B) shows Sn (1 atom) doped ZnO super cell, Fig 1(C) shows Sn (2 atom) doped ZnO supercell and Fig 1(D) shows Sn (3 atom) doped ZnO supercell.

## Results and discussion

The optical properties of Sn doped ZnO crystal, such as electronic bandgap structure, extinction coefficient (k), refractive index (n), absorption coefficient, and reflectance, are calculated using DFT and the CASTEP toolkit package of Material Studio software. The ZnO super cells (imported from material studio's inorganic crystal structure database) are displayed in Fig 1 (A). As shown in Fig 1A–1D, Sn atoms were interstitially integrated in ZnO crystal. Zinc (Zn) atoms are represented by grey coloured balls, oxygen (O) atoms are represented by red balls, and tin (Sn) dopant atoms present at interstitial sites of ZnO crystal are represented by green coloured balls. As indicated in Fig 1(B), several interstitial Sn atoms have been introduced in the ZnO super cell at its lattice positions to proportionally enhance the Sn concentration. Similarly, the Sn content of ZnO super cell at its lattice sites is enhanced further by the

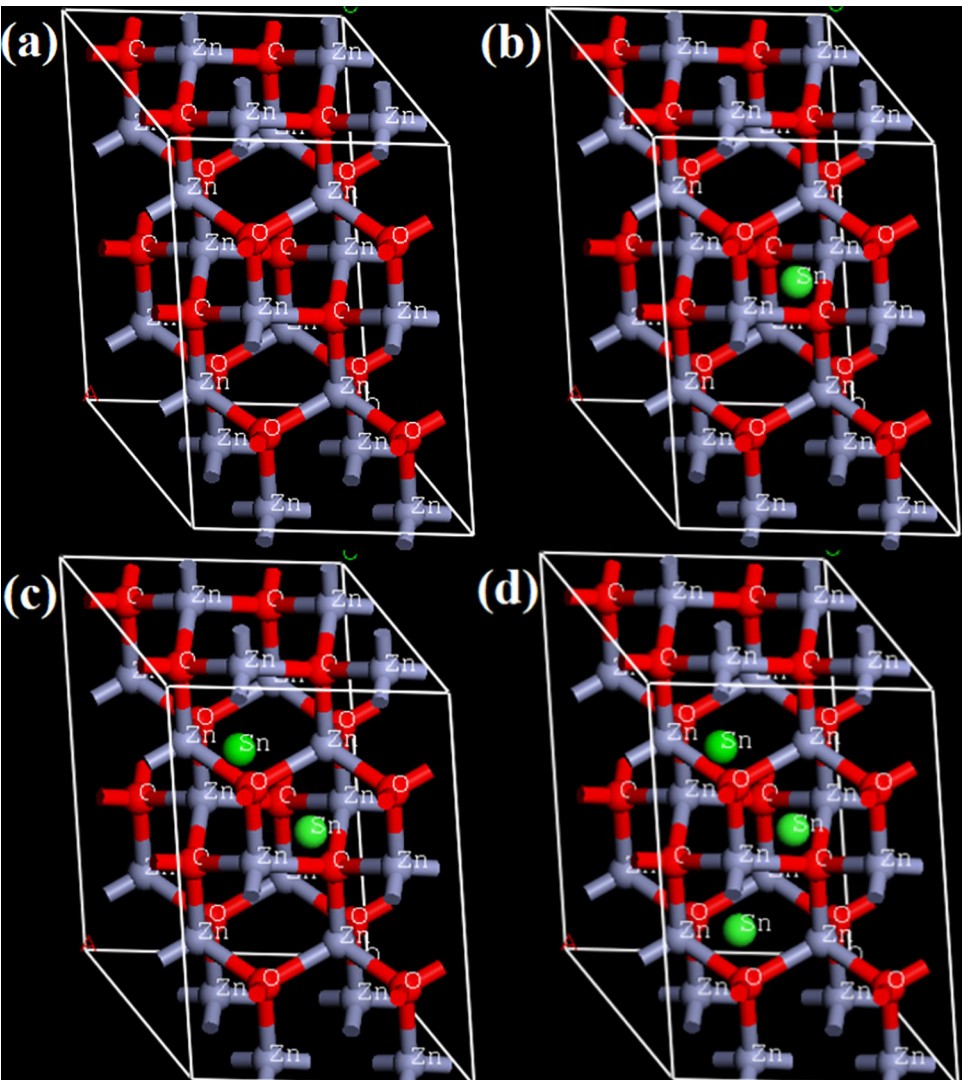

**Fig 1. Crystal structure of undoped and doped, 2×2×2 ZnO supercell.**

incorporation of some additional Sn atoms, as illustrated in Fig 1C and 1D, which successively demonstrates a considerably greater number of interstitial Sn atoms in ZnO super cell than Fig 1(A) and 1(B). Thus, Fig 1(A) depicts an undoped ZnO supercell, Fig 1(B) depicts a Sn (1 atom) doped ZnO supercell, Fig 1(C) depicts a Sn (2 atom) doped ZnO supercell, and Fig 1(D) depicts a Sn (3 atom) doped ZnO supercell.

To begin with, the electronic bandgap structures of Sn doped ZnO are obtained by first principle calculation using density functional theory (DFT) which is implemented using CASTEP tool kit of Material Studio simulation software. The basic equations which governs the electronic band structure in CASTEP tool kit are expressed by Eq (1), Eq (2) and Eq (3). Bloch's theorem emphasizes that in a periodic or regular system, each electronic wave function may be expressed as a product of a cell-periodic portion and a wavelike part [24]. These equations impose periodic boundary conditions that pertain to this theorem.

$$\varphi_i(r) = e^{ikR}\varphi_i(r) \tag{1}$$

Where $\varphi$ is pseudo wave function, $k$ is wave vector, $R$ is displacement function and $r$ is radial distance in spherical coordinate system for a point in space.

A discrete collection of plane waves whose wave vectors are reciprocal lattice vectors of the crystal can be used as a basis set to expand the cell periodic component, $\varphi$.

$$\varphi_i(r) = \sum C_{i,G} e^{iGR} \tag{2}$$

Where $\varphi$ is pseudo wave function, $k$ is wave vector, $G$ is unique constant density function, $R$ is displacement function and $r$ is radial distance in spherical coordinate system for a point in space.

Thus, the total energy of all electronic functions is expressed as the sum of plane waves, exp [i(k+G)R]. The Fourier transforms of the potentials (electron-ion, Hartree, exchange-correlation) are used to characterise the shape, and the kinetic energy is diagonal. In addition to facilitating efficient geometry optimization and molecular dynamics techniques, the plane wave basis set makes it simple to compute derivatives of the total energy with regard to atomic displacements (that is, stresses and forces are computationally inexpensive in this approach). Convergence analysis and optimization for the plane-wave basis set are very simple tasks because of the methodical approach to incorporating more basis functions [25].

$$\sum [|k + G|^2 \delta_{GG'} + V_{ion}(G - G') + V_{xc}(G - G')] + C_{i,k+G'} = \varepsilon_i C_{i,k+G} \tag{3}$$

Where $\varphi$ is pseudo wave function, $\delta_{GG}$ Kronecker delta function, $k$ is wave vector, $G$ is unique constant density function, $V_{xc}$ is exchange–correlation potential, $V_{ion}$ is ionic potential and $r$ is radial distance in spherical coordinate system for a point in space. From Fig 2A–2D, it is observed that of band gap values of Sn doped ZnO reduces as the Sn dopant atoms increases in ZnO crystal. This is mainly due to band gap narrowing effect which is attributed mainly due to +4 oxidation sate of Sn atoms, as a result there is enhancement in number of electron carriers concentration in Sn doped ZnO crystal due to which effective band gap of Sn doped ZnO crystal reduces [26]. Therefore, with increase in Sn doping in ZnO crystal, the value of band gap decreases. Thus, bandgap values of undoped ZnO was obtained to be 3.2 eV, bandgap of Sn (1 atom) doped ZnO was found out to be 2.8 eV, and bandgap of Sn (2 atoms) and Sn (3 atoms) doped ZnO were found out be 1.8 eV and 1.4 eV, respectively.

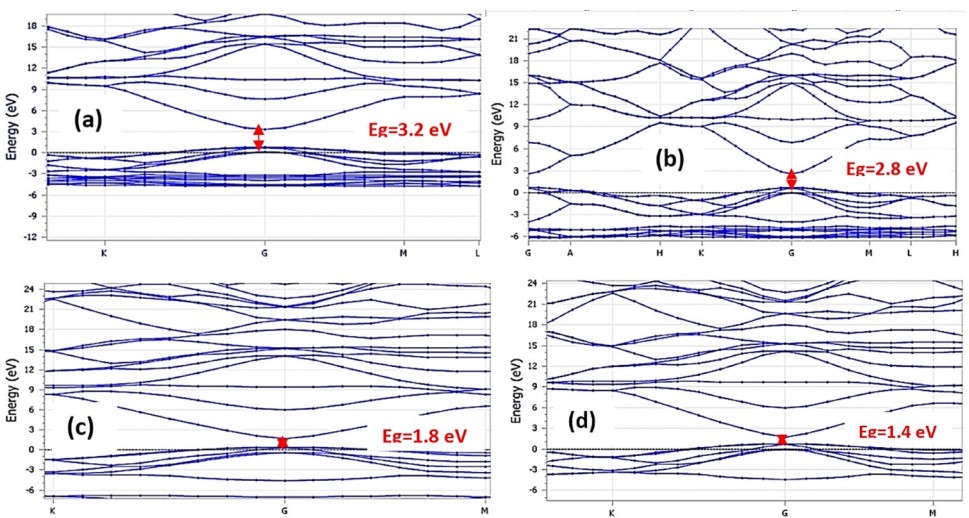

**Fig 2. Electronic band structures of undoped and doped, 2×2×2 ZnO supercell.**

Furthermore, Fig 3(A) shows the reflectance characteristics of Sn doped ZnO for various amount of Sn dopants in ZnO crystal. It is observed that the reflectance properties of Sn doped ZnO increases as the Sn dopants atoms increases in ZnO crystal. This is mainly because increase in doping of Sn metal atoms in ZnO crystal, due to which light get more reflected by these Sn metal atoms. Also, the trend of extinction coefficient and absorption coefficient are obtained theoretically. It is observed from Fig 3B and 3C that firstly there is an increment in extinction coefficient (k) and absorption coefficient (α) of Sn doped ZnO crystal with increase in Sn dopants atoms in ZnO crystal, but at higher concentration of Sn dopant atoms in ZnO crystal, the extinction coefficient and absorption coefficient decreases. This can be explained by considering the combined effect of band gap and reflectance of Sn doped ZnO crystal. As Sn doping is increased, the band gap value Sn doped ZnO crystal reduces or become narrower due to which light absorption capability increases over wider range of wavelength of solar spectrum, consequently there is an increment in absorption coefficient or extinction coefficient with Sn atoms doping, but at the same time reflectance of Sn doped ZnO also increases

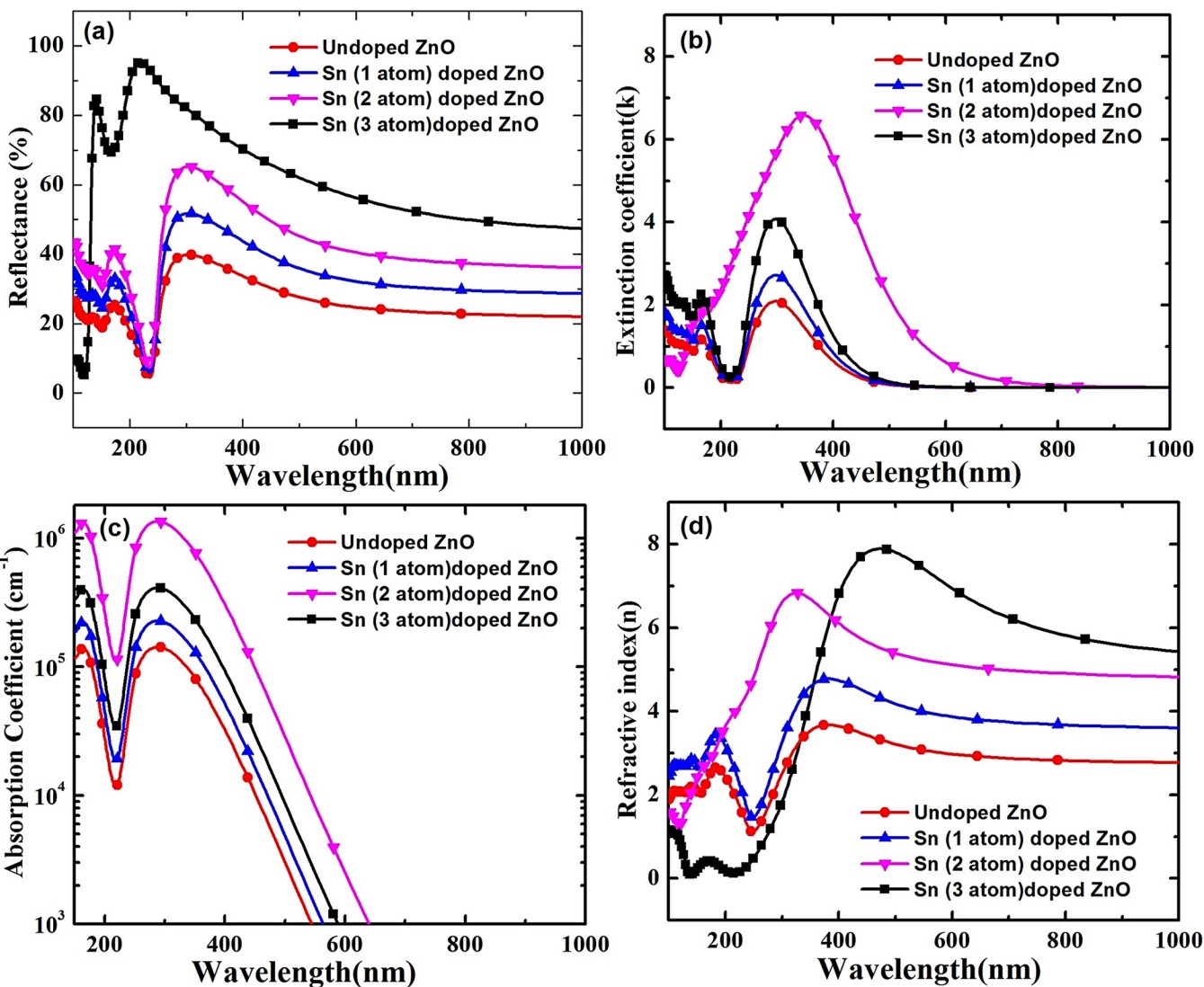

Fig 3. Reflectance, extinction coefficient, absorption coefficient and refractive index of undoped and doped, 2×2×2 ZnO supercell.

with an increase in Sn doping. At very high Sn doping reflectance is dominated, therefore extinction coefficient and absorption coefficient begin to decrease due to trade-off between band gap value (responsible for light absorption) and reflectance properties of Sn doped ZnO crystal with increase in Sn doping. Finally, at particular value of Sn doping they counter balance each other and gives an optimum value of extinction and absorption coefficient. So, Sn (2 atom) doped ZnO crystal shows highest extinction and absorption coefficient as shown in Fig 3(B) and 3(C).

Some basic equations which governs the optical properties of thin film such as refractive index along with extinction coefficient, reflectance and absorption coefficient are mentioned below as Eq (4), Eq (5) and Eq (6), respectively [27]. The Eq (4) represents the complex refractive index (N), which is actually sum of real refractive index (n) and extinction coefficient (k). The total reflectance (R) of light from a thin film is calculated by using Eq (5), whereas the absorption coefficient (α) indicates the portion of photon or light energy lost by the light wave when it traverse through the material, therefore the photon or light intensity I(x) at the distance x from the material surface is denoted by Eq (6).

$$N = n + ik \tag{4}$$

$$R = \frac{|1 - N|^2}{|1 + N|^2} = \frac{(n-1)^2 + k^2}{(n+1)^2 + k^2} \tag{5}$$

$$I(x) = I(0)\exp(-\alpha x) \tag{6}$$

$$\text{Where, } \alpha = \frac{2k\omega}{c} \tag{7}$$

We extended our study further to investigate the wavelength dependent refractive index (n) of Sn doped ZnO crystal which is shown in Fig 3(D). It is observed that refractive index of Sn doped ZnO increases with an increase in metallic Sn doping, and the results are in good agreement with refractive index values reported in literatures [28]. The undoped ZnO shows the least refractive index, whereas as the Sn (3 atom) doped ZnO crystal shows the highest refractive index. This is mainly due to the fact that dispersion energy and oscillator energy values of the ZnO are enhanced with increase in metallic dopants such as chromium (Cr), cobalt (Co), tin (Sn), etc. [29], consequently, resulting into higher refractive with an increase in Sn doping as per Wemple–DiDomenico (WDD) dispersion relationship, which is expressed by Eq (8) [30].

$$n^2 - 1 = \frac{E_0 E_d}{E_0 - \hbar^2 \omega^2} \tag{8}$$

Where n is refractive index, $E_0$ and $E_d$ are oscillator and dispersion energies, respectively.

Furthermore, in order to observe the structural property of Sn doped ZnO crystal, XRD patterns were obtained by XRD tool kit of material studio as shown in Fig 4A–4D. The Fig 4 (A) shows the XRD patterns of undoped ZnO crystal, Fig 4(B) shows the XRD patterns of Sn (1 atom) doped ZnO, Fig 4(C) shows the XRD patterns of Sn (2 atom) doped ZnO and Fig 4 (D) shows the XRD patterns of Sn (3 atom) doped ZnO. From XRD patterns it is observed that as Sn doping is increasing, consequently there is increment in number of XRD peaks which indicates that ZnO is being converted from crystalline to multi crystalline nature which may results in some unwanted recombination such as grain boundary recombination, interface

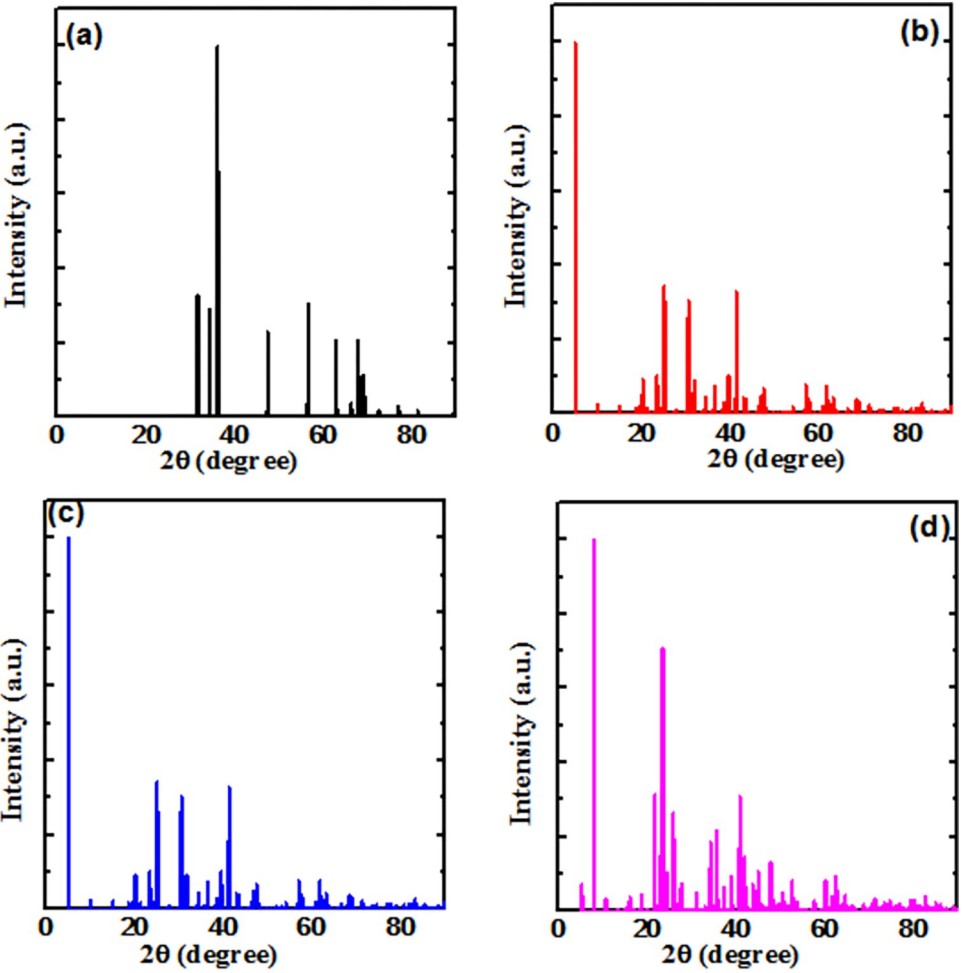

**Fig 4. XRD patterns of undoped and doped, 2×2×2 ZnO supercell.**

traps recombination [23]. Hence, Sn doping must be regulated properly to get an optimized structural property.

Finally, comparing the aforementioned structural and optical properties of Sn doped ZnO, it has been observed that obtained electronic bandgap values (1.4 to 3.2 eV) and wavelength dependent refractive index are in good agreement with values reported in literatures [31, 32]. However, the wavelength dependent extinction coefficient and absorption coefficient of Sn doped ZnO shows a slight variation in the wavelength of 300 nm to 600 nm wavelength. This variation may be result of change in Sn-dopant concentrations in ZnO crystal. Moreover, some important properties of Sn doped ZnO has been summarized in Table 1 for ready references.

## Conclusion

This work can concluded by stating that electronic bandgap structures and band gap values of Sn doped ZnO can be tuned by varying Sn dopants. It is also observed that band gap reduces as the Sn dopants atoms increases in ZnO crystal. One of the key reasons for reduced band gap would be band gap narrowing effect due higher carrier (electron) concentration. It is well known fact that narrower band gap enhance the probability of light absorption over wider

**Table 1. Different band gap values of Sn doped ZnO along with its refractive index (n (λ = 0)), extinction coefficient (k (λ = 0)), absorption coefficient (α (λ = 0)), and reflectance, R ((λ = 0)).**

| Parameters | Undoped ZnO | Sn (1 atom) doped ZnO | Sn (2 atom) doped ZnO | Sn (3 atom) doped ZnO |
|---|---|---|---|---|
| Band gap (eV) | 3.2 | 2.8 | 1.8 | 1.4 |
| Refractive index [n(0)] | 2 | 3 | 1.8 | 1.5 |
| Extinction coefficient, [k(0)] | 1 | 2 | 0.8 | 3 |
| Absorption Coefficient, $\alpha(0)$, [cm$^{-1}$] | $2\times10^5$ | $4\times10^5$ | $2\times10^6$ | $6\times10^5$ |
| Reflectance, R(0), [%] | 30 | 25 | 35 | 40 |

range of wavelength. Further, the trend of extinction coefficient and absorption coefficient are calculated. It is observed that there is an increase in extinction coefficient and absorption coefficient of Sn doped ZnO crystal with increase in Sn dopants atoms in ZnO crystal, but at higher Sn dopant atom in ZnO crystal the extinction coefficient and absorption coefficient decreases. This study concludes that Sn doped ZnO films with suitable light absorption property or extinction coefficient in the visible region of the solar spectrum and the potential of band gap engineering may be a desirable option of material for photosensitive devices. The investigation of tunable optical and electronic bandgap structure of Sn doped ZnO crystal will open the way for a wide range of photosensitive devices.

## Supporting information

**S1 Data.**
(RAR)

## Acknowledgments

The authors would like to thank faculties and staff members of Department of Mechanical Engineering and Chemical & Materials Engineering, Adama Science and Technology University, Ethiopia, for their unconditional support.

## Author Contributions

**Conceptualization:** Manoj Kumar.

**Data curation:** Bittu Kumar.

**Formal analysis:** Purnendu Shekhar Pandey.

**Funding acquisition:** Gyanendra Kumar Singh.

**Investigation:** Gyanendra Kumar Singh.

**Methodology:** Banoth Ravi.

**Supervision:** S. V. S. Prasad.

**Validation:** Santosh Kumar Choudhary.

**Writing – review & editing:** Rajesh Singh.

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
