## [Decision Letter · Decision Letter 0]

13 Nov 2023

PONE-D-23-35532Impact of Sn-Doping on the Optoelectronic Properties of Zinc Oxide Crystal: DFT ApproachPLOS ONE

Dear Dr. Singh,

Thank you for submitting your manuscript to PLOS ONE. After careful consideration, we feel that it has merit but does not fully meet PLOS ONE’s publication criteria as it currently stands. Therefore, we invite you to submit a revised version of the manuscript that addresses the points raised during the review process.

We look forward to receiving your revised manuscript.

Kind regards,

A M Mansour, Ph.D.

Academic Editor

PLOS ONE

Journal Requirements:

4. Thank you for stating the following financial disclosure: "This work is supported by Adama Science and Technology University, Adama, Ethiopia".

5. Thank you for stating the following in the Acknowledgments Section of your manuscript: "This work is supported by Adama Science and Technology University, Adama, Ethiopia."

Please remove any funding-related text from the manuscript and let us know how you would like to update your Funding Statement. Currently, your Funding Statement reads as follows: "This work is supported by Adama Science and Technology University, Adama, Ethiopia"

7. PLOS requires an ORCID iD for the corresponding author in Editorial Manager on papers submitted after December 6th, 2016. Please ensure that you have an ORCID iD and that it is validated in Editorial Manager. To do this, go to ‘Update my Information’ (in the upper left-hand corner of the main menu), and click on the Fetch/Validate link next to the ORCID field. This will take you to the ORCID site and allow you to create a new iD or authenticate a pre-existing iD in Editorial Manager. Please see the following video for instructions on linking an ORCID iD to your Editorial Manager account: https://www.youtube.com/watch?v=_xcclfuvtxQ

Reviewers' comments:

Reviewer's Responses to Questions

**Comments to the Author**

1. Is the manuscript technically sound, and do the data support the conclusions?

Reviewer #1: Yes

Reviewer #2: Partly

2. Has the statistical analysis been performed appropriately and rigorously? 

Reviewer #1: N/A

Reviewer #2: Yes

3. Have the authors made all data underlying the findings in their manuscript fully available?

Reviewer #1: No

Reviewer #2: Yes

4. Is the manuscript presented in an intelligible fashion and written in standard English?

Reviewer #1: No

Reviewer #2: Yes

5. Review Comments to the Author

Reviewer #1: Dear Editor of Journal PLOS ONE.

Thanks for sending me this manuscript (Impact of Sn-Doping on the Optoelectronic Properties of Zinc Oxide Crystal: DFT Approach) for review.

The prepared materials were characterized using First principal calculation using density functional theory (DFT) of Sn doped ZnO crystal has been performed to analyze the effect of Sn interstitial doping in ZnO on its optoelectronic properties.

1. The title is appropriate for the content.

the abstract is not convenient and should be modified to focus on the obtained results. - It's better to change the title with remove the word: Basically, out,…

2. Rewrite the abstract: indicate the most interesting results, avoid repetitions and check the English grammar.

3. The novelty of the work needs to be declared in the introduction part.

4. The whole manuscript should be revised for grammatical and writing errors.

5. The introduction and results are somewhat interesting but needs to be specific and focused on the finding and work novelty.

6. Authors must compare their results with the previously reported.

Finally, after a minor revision, the manuscript can be accepted for publication in the journal.

Reviewer #2: Comments

Dear Editor of Journal PLOS ONE.

This manuscript reported the Impact of Sn-Doping on the Optoelectronic Properties of Zinc

Oxide Crystal: DFT Approach. However, the manuscript is well written and the results are not explained properly. Hence, the manuscript needs more modifications in the present state. Therefore, a major revision should be made before its publication.

Other comments:

-The Abstract is poor language so authors must rewrite.

-Replace First to the first.

-Delete Basically,

-Put and before absorption coefficient;

-detailed out in this study. Delete out

-doping in ZnO play replace to : doping in ZnO plays

-Authors should include their finding in the abstract

Kindly, support the application of ZnO-based in scientific and industrial fields. Refe..., J. Environ. Manage. 270, 110816 (2020); ECS J. Solid State Sci. Technol. 10, 063007 (2021),; Int. J. Environ. Sci. Technol. 17, 4481–4494 (2020); https://doi.org/10.1007/s12633-022-01886-2

https://doi.org/10.1149/2162-8777/ac07f9

https://doi.org/10.1007/s12633-021-01618-y

https://doi.org/10.1007/s10854-022-08158-0

10.21608/EJCHEM.2021.87118.4214

https://doi.org/10.1088/1402-4896/ac119e

https://doi.org/10.1007/s10854-021-07182-w

- Authors must give reasons for their finding (The undoped ZnO show the least refractive index, whereas as the Sn (3 atom) doped ZnO crystal shows the highest refractive index)?

- Check the English grammar of the text.

Regards

6. PLOS authors have the option to publish the peer review history of their article (what does this mean?). If published, this will include your full peer review and any attached files.

Reviewer #1: No

Reviewer #2: No

---

## [Author Response · Author response to Decision Letter 0]

29 Nov 2023

Dear Editor,

Thank you for allowing a resubmission of our manuscript, with an opportunity to address the reviewers’ comments.

We are uploading (a) our point-by-point response to the reviewer comments (response to reviewers), (b) an updated revised manuscript with yellow highlighting indicating changes, and (c) a clean updated manuscript without highlights (Main Manuscript).

Kindly accept our revised manuscript for further processing.

---

## [Editor Report · Decision Letter 1]

5 Dec 2023

Impact of Sn-Doping on the Optoelectronic Properties of Zinc Oxide Crystal: DFT Approach

PONE-D-23-35532R1

Dear Dr. Singh,

We’re pleased to inform you that your manuscript has been judged scientifically suitable for publication and will be formally accepted for publication once it meets all outstanding technical requirements.

Kind regards,

A M Mansour, Ph.D.

Academic Editor

PLOS ONE

Additional Editor Comments (optional):

accept
---

## [Editor Report · Acceptance letter]

19 Dec 2023

PONE-D-23-35532R1 

PLOS ONE

Dear Dr. Singh, 

I'm pleased to inform you that your manuscript has been deemed suitable for publication in PLOS ONE. Congratulations! Your manuscript is now being handed over to our production team.

Kind regards, 

on behalf of

Prof A M Mansour 

Academic Editor

PLOS ONE